# Is the Intrasexual Competition in Male Red Deer Reflected in the Ratio of Stable Isotopes of Carbon and Nitrogen in Faeces?

**DOI:** 10.3390/ani13142397

**Published:** 2023-07-24

**Authors:** Giovanni Vedel, Eva de la Peña, Jose Manuel Moreno-Rojas, Juan Carranza

**Affiliations:** 1Wildlife Research Unit (UiRCP), University of Córdoba, 14071 Córdoba, Spain; giove985@gmail.com (G.V.); jcarranza@uco.es (J.C.); 2Instituto de Investigación en Recursos Cinegéticos, IREC (CSIC, UCLM, JCCM), Ronda de Toledo 12, 13005 Ciudad Real, Spain; 3Department of Agroindustry and Food Quality, Andalusian Institute of Agricultural and Fisheries Research and Training (IFAPA), Alameda del Obispo. Avda. Menéndez Pidal, s/n., 14071 Córdoba, Spain; josem.moreno.rojas@juntadeandalucia.es

**Keywords:** male-male competition, isotopes analyses, faeces, male red deer, nutritional stress

## Abstract

**Simple Summary:**

Stable isotope analysis is a technique increasingly used for studying many aspects of wildlife behaviour, such as diet, migrations, and trophic relationships. However, scarce information is available on how isotopes are assimilated into the body under conditions such as nutritional stress situations or intrasexual competition for mates. In this study, we evaluated how intrasexual competition between male red deer from free-ranging populations in Andalusia (Spain) is reflected in the ratio of stable isotopes of carbon and nitrogen (δ^13^C and δ^15^N) in faeces. Our results showed that in populations with high intrasexual competition δ^15^N values in faeces were significantly lower than in low intrasexual competition populations. No differences were found for δ^13^C. This study provides a basis for interpreting δ^15^N values related to different protein usage under various nutritional stress situations in male red deer and novel applications of the use of nitrogen isotopes.

**Abstract:**

Isotopic analysis of carbon and nitrogen in faeces is a reliable methodology for studying ecology in wildlife. Here, we tested this technique to detect variations in carbon and nitrogen isotopic ratios (δ^13^C and δ^15^N) in two different intrasexual competition scenarios of male Iberian red deer (*Cervus elaphus hispanicus*) using faeces of individuals collected during hunting actions in South-eastern Spain. The carbon isotopic ratio (δ^13^C) was not found to be significant, likely due to similar diet composition in all individuals. However, the nitrogen isotopic ratio (δ^15^N) was found to be lower in populations where sexual competition between males during the rut was higher compared to low-competition populations. Therefore, this study suggests a different use of proteins by an individual male red deer depending on the sexually competitive context in which he lives. Although further research is needed, these results show the potential of isotopic analysis as a tool for studying individual and populational variations in the level of intrasexual competition, with implications in evolutionary ecology and population management.

## 1. Introduction

Intrasexual competition is the process by which individuals of the same sex compete for access to potential mates [1]. In polygynous species, females are a limiting resource for which males compete [2]. During intrasexual competition, males use a wide repertoire of behaviours including vocal challenges, fights and displaying of sexual traits that contribute to arranging dominance relationships and access to mates [1,2,3]. However, most sexual behaviours and traits are costly and plastic in their expression, so that males can modulate them according to the level of sexual competition in the population [4].

The red deer is a polygynous species and a good model for sexual selection and intrasexual competition studies [5,6,7,8]. It has been shown for this species that in cases where there are few competitors, and hence there is not a strong pressure to obtain matings, males may reduce their allocation to sexual behaviour and sexual traits [9,10,11,12,13,14]. 

Male–male competition affects feeding behaviour (e.g., tooth wear, [10]), but also, we may expect differences in nutrient metabolism. Understanding how molecules are distributed and metabolised by the organism in different scenarios in which an individual can live is of increasing interest in ecology [15,16]. The quantification and ratio of naturally occurring stable isotopes of carbon and nitrogen (δ^13^C and δ^15^N) in the food and tissues of those who consume it is a key tool for understanding the diet of animals [17,18,19]. This requires consideration of the isotopic fractionation that occurs during primary production processes and throughout the nitrogen cycle [20,21]. The effects of isotopic fractionation on primary food sources are modified by metabolic processes in animals and, ultimately, incorporated into tissues or excreted as residues [22]. 

δ^13^C values reflect the type of vegetation consumed by an organism [19]. Plants have different photosynthetic pathways, and this has consequences for different δ^13^C ratios in plant tissues [23]. As a result, the δ^13^C value of an organism reflects the proportion of the δ^13^C value of the diet [24]. This has been used to infer changes in diet [25] and has been proposed as a key tool for comparing the diet of individuals of the same species under different management conditions [19]. On the other hand, the δ^15^N values are determined by multiple factors. One main feature of nitrogen-stable isotopes is their relationship with the trophic level of an individual due to the predictable increase in δ^15^N value (usually between 2‰ and 4‰ per trophic level) caused by the trophic discrimination factor that occurs along the food chain [18,26,27,28,29], as it varies depending on the quantity and quality of proteins in the diet [29]. But also, the δ^15^N values show different ranges for tissues in the organism and faeces due to the deamination process [23]. Deamination enzymes preferentially remove amino groups with ^14^N. Therefore, the nitrogen excreted is enriched with ^14^N compared to the tissue of animals [30]. The δ^15^N values can also indicate whether an individual is in a situation of food and/or health stress since, when an individual does not ingest enough proteins, the body can resort to the catabolism of their own tissues. Therefore, the body tissues used to make up for the lack of protein intake repeat the same isotope fractionation processes leaving the body tissues of a nutritionally stressed individual more enriched in δ^15^N [31,32,33,34,35]. In this context, the presence of parasites in organisms is of relevance since it is well known that parasites have a detrimental effect on the body condition [36,37], immune response and behaviour of their hosts [38]. Parasitisation leads to isotopic variations that influence physiological processes related to isotope fractionation and the distribution of stable isotopes in the organism [34]. Previous studies have shown that parasitism increases δ^15^N values in tissues [39,40].

During the rut, male red deer in Iberia suffer great nutritional stress for mainly two reasons. The first is that it is a time of food scarcity due to the summer drought that precedes the mating season [6,36,41,42]. Additionally, during this period, males are involved in strong intrasexual competition, when individual energy is allocated towards defending groups of females or mating territories [7]. Poor nutritional status and nutritional stress should influence the physiological processes underlying the fractionation and distribution of isotopes in the body [35]. 

The Iberian red deer (*Cervus elaphus hispanicus*) in southern Spain mostly occurs in hunting areas. In this region, the types of management of red deer populations have been shown to affect population structure (age and sex) with consequences on ecological and evolutionary processes and in the degree of intrasexual competition among males [9,10,12,13,43,44,45]. Population structure and the associated degree of competition to which an individual is subjected have notable impacts on behaviour [43,44], morphology [10] and health [12].

The main goal of this work was to investigate the effect of the level of male intrasexual competition in the population on the faecal values of stable isotopes of carbon and nitrogen in male red deer. Specifically, we hypothesised potential differences in nitrogen isotopic values between males in populations with high and low levels of intrasexual competition, while carbon isotopes may remain similar if vegetation on which animals feed in both types of areas is similar. Additionally, we wanted to explore the effects of different morphological variables on isotope values and physical conditions, including the presence of parasite load.

## 2. Materials and Methods

### 2.1. Study Area and Populations

We investigated male Iberian red deer harvested in hunting estates in southwestern Spain (Andalusia and Extremadura regions). This study was carried out in nine populations concretely located in Sierra Morena (Province of Córdoba, UTM 37°58′ N, 5°05′ W) and Sierra San Pedro (Province of Cáceres, UTM 39°19′ N 6°42′ W). The vegetation present is similar throughout the study zone. It is characterised by mountain ranges covered by open oak agroforestry woodland, known as *dehesa*, accompanied by Mediterranean species such as *Cistus* spp., *Erica* spp., *Arbutus unedo*, *Phyllirea* spp., *Genista hirsuta*, *Lavandula* spp. and *Olea europaea* [12,13,40]. Each hunting estate constitutes a different population because of the existing natural (i.e., mountains, geographical distance) and artificial (i.e., fences) barriers between them. The average estates area in our study area was 2347 ha, being a minimum area of 503.07 ha and a maximum area of 9223.18 ha. Two different management regimes apply to hunting estates in our study area. In one regime, estates are fenced with stock-proof wire mesh, and red deer are not allowed to move across nearby estates, while in the other regime, estates are unfenced [43,44]. In unfenced estates, the free movement of animals from one estate to another means that as many as possible male red deer are hunted in each estate to prevent them from being hunted in neighbouring estates [44]. This results in a sex ratio bias toward females. In this scenario, males can mate without the need to compete with other rivals [10,43,44] has been shown via genetic tools [8,44]. On the contrary, fenced estates maintain more equilibrated age and sex population structures that allow intrasexual competition to operate more naturally [43]. In unfenced estates, the average area was 3948.44 ha, while in fenced estates, the average area was 1074.28 ha. Females’ availability is lower at these estates, and males must compete to mate with them [10,43,44]. There is no difference between unfenced and fenced estates in terms of population density due to these two management situations [44], the density of individuals in both types of populations does not differ [44], being around 0.3 individual/ha (0.1–1.0 indiv. /ha) [44]. However, they show contrasting scenarios of male intrasexual competition for mates, referred to as high competition (HC) in fenced estates and low competition (LC) in unfenced estates in several previous studies [9,10,11,12,13,45]. Specifically, previous work has reported that the sex ratio in open hunting estates was mean ± SE = 4.76 ± 0.40 (females/males), while in fenced hunting estates, the sex ratio was mean ± SE = 1.58 ± 0.13 (females/males) [43,44]. Henceforth, we will refer to fenced estate populations as having high mate competition level (HC) and to unfenced estate populations as having low mate competition level (LC).

### 2.2. Sample Collection

Seventy-eight male Iberian red deer were sampled from nine different hunting estates (forty-six individuals from fenced hunting estates and thirty-two males from unfenced hunting estates) during the 2017–2018 hunting season (from the 15 October to 15 February).

Morphological measurements were taken of harvested individuals a few hours after the shot. We measured body head–tail length (in cm, BL), thoracic perimeter (in cm, TP) and the maximum length of the dark ventral patch (in cm, DVP; see the methods in Carranza et al., 2020b). The average antler length (in cm, AL) was calculated by measuring from the lowest outer edge of the burr on the outer side to the most distant point of the main beam. Age was estimated by counting the marks on the inter radicular pad under the first molar [46]. 

Faeces were collected from the rectum of each animal for parasitological analyses and to determine the diet composition by the stable isotopes’ technique. Faecal samples for stable isotope analyses were frozen at—20 °C until laboratory analyses. A subsample of fresh faecal samples was used to coprological analyses in the Parasitology Department of the Veterinary Faculty of the University of Córdoba (Córdoba, Spain) following the methods in [12]. Sixty-seven faecal samples were collected from seven populations (four unfenced estates and three fenced estates). A total of forty-two males were sampled from forty-six individuals in HC populations for this study. From LC populations, samples were collected from twenty-five males out of thirty-two (see Appendix A). The individuals included in this study were examined for the presence or absence (i.e., prevalence) of gastrointestinal nematodes, trematodes, cestodes, coccidian oocysts, and bronchopulmonary nematode larvae. However, only gastrointestinal parasites (*Elaphostrongylus cervi*) and bronchopulmonary parasites (*Strongylida* spp.) were found to be infected. Therefore, the presence/absence of parasite prevalence refers to both parasite groups. 

### 2.3. Stable Isotope Ratio Analyses

Faecal samples were dried and lyophilised individually, then ground to obtain a fine and homogeneous powder. The resulting compound was placed in tin capsules to obtain a weight between 700 and 1000 micrograms using a precision microbalance (XP6 METTLER TOLEDO, 0.1 UG). Each capsule was then analysed individually to obtain the carbon and nitrogen isotopic ratios using an isotopic ratio mass spectrometer (IRMS, Delta V Advantage*) equipped with an elemental analyser (EA; Flash 2000 HT*) and a universal interface for continuous flow analysis (Con Flow IV*). *Thermo Fisher Scientific (Bremen, Germany). 

The results of the analyses are reported in parts per thousand (‰) relative to V-PDB (Vienna Pee Dee Belemnite) and Air Atmospheric following the calculation of the standard notation δ. International standards IAEA-CH7-polyethylene (δ^13^C = −32.15‰) and IAEACH6-sucrose (δ^13^C = −10.4‰) were used to calibrate the CO_2_ gas cylinder (used for δ^13^C measurements). IAEA-N1-ammonium sulphate (δ^15^N = 0.4‰) and IAEA-N2-ammonium sulphate (δ^15^N = 20.3‰) were used for the calibration of the N_2_ gas cylinder (used for δ^15^N measurements). International standards USGS-40 (δ^13^C = −26.24‰ and δ^15^N = −4.52‰), USGS-42 (δ^13^C = −21.09‰ and δ^15^N = +8.05‰) NBS-22 (δ^13^C = −29.72‰) and secondary standards LIE-PA (δ^13^C = −15.77‰ and δ^15^N16.55‰) and protein-IVA (δ^13^C = −26.98‰ and δ^15^N = 5.94‰) were repeatedly analysed at intervals to check for possible drift corrections. The precision of measurements was better than (0.2‰) (one standard deviation) for both elements.

### 2.4. Statistical Analyses

We used linear mixed-effects regression models (LMMs) to predict the responses of δ^13^C and δ^15^N against several predictors (parasitised (Yes/No), age, age^2^, antler length, body length, thoracic perimeter, and the dark ventral patch size), and their meaningful interactions, in LC and HC populations and controlling for month (October to February) as random source of variation. The statistical analyses were performed using data from those individuals from whom information was available for all the variables to be tested (N = 67).

Using Shapiro–Wilk test, we checked for both normality and homoscedasticity of the models’ residuals when explaining variation in δ^13^C and δ^15^N. We also checked the presence of outliers and singularity of both models. We z-transformed all the quantitative variables to facilitate model convergence using the scale function in R. We also calculated the variance inflation factors (VIFs; Alin 2010) of each model using the package *usdm* [47] to avoid multicollinearity between variables (VIF < 2). We set statistical significance as *p* < 0.05.

To avoid risks of over-parameterisation, full models were simplified by backward elimination, removing non-significant (*p*-value > 0.05) interaction terms one at a time, following the marginality principle (i.e., higher-order interactions were tested first and, if significant, then lower-order effects). Backward elimination was based on *p*-values in favour of information theory approaches such as ΔAIC [48].

We calculated the coefficients of the final models using the REML-restricted maximum likelihood method [49]. The variance explained by the models was represented as marginal R^2^ (variance explained by fixed effects) and conditional R^2^ (variance explained by random and fixed effects) following a method developed for linear mixed-effects models [50].

All analyses and graphics were conducted in R v.3.6.1 (R Foundation for Statistical Computing, Vienna, Austria) using the package *lme4* [51] and *ggeffects* [52] packages.

## 3. Results

In the case of δ^13^C values, we did not find a significant effect of any of the tested variables (Table 1). Neither male characteristics (dark ventral patch size, antler length, body length, thoracic perimeter, age, and age^2^) nor health status (be parasitised or not) nor the type of population (HC or LC) had a significant effect on the δ^13^C values. For the δ^15^N data, we detected an effect of both types of populations (HC or LC) and the presence or absence of parasites in individuals (Table 2). In HC populations, δ^15^N values were significantly lower than in LC populations. However, parasitised individuals exhibited lower δ^15^N values than unparasitised individuals in both types of populations (Figure 1). Appendix A summarises the descriptive statistics for the variables used as covariates included in LMM1 and LMM2 by population type (HC vs. LC). 

## 4. Discussion

The stable isotopes not only provide dietary information but also offer insights into the physiological metabolism of animals [31,35]. The δ^13^C values can reveal the type of plant taxa an animal favours in the case of herbivores [53], whereas δ^15^N value is closely related to protein intake [54].

The main result of this work is the difference in δ^15^N value between Iberian red deer males in HC and LC populations in the South of Spain. Individuals from populations with high levels of male-male competition (HC) had significantly lower δ^15^N values than those from LC populations. However, no differences between both populations have been shown regarding the δ^13^C value, indicating that the type of diet was similar. Even feeding on similar types of plants, the metabolism of N, as indicated by δ^15^N values, differed. Sampling for this study was carried out during the red deer hunting season, just after or even ending the rut in this species. The mating season is energetically costly for males [55] and usually implies a rapid drop in body weight during the rut and the consequent deterioration in individual body conditions [56]. Depending on the individual and social contexts (i.e., the number of competitors that produce different situations of HC and LC), individuals may use different strategies to cope with energy demands [57]. Male red deer behaves mostly as capital breeder [58]; they eat very little during the rutting season, with energy for the rut coming mainly from reserves accumulated during the previous spring and summer, stored as body fat [5,59]

Our results suggest that the differences found in δ^15^N value may be related to the differential effort and energy expenditure during the rutting season of males on HC and LC estates. Previous studies have shown differences in male–male competition effort and in the development of sexual traits in both types of populations [9,10,11,12,13,14,45]. Thus, this work suggests that these two levels of competition (HC and LC) may also have consequences in male red deer metabolism, revealed by the δ^15^N value. 

Isotopic values in this study have shown that, in HC estates, a higher proportion of ^14^N compared to ^15^N was excreted in the faeces, with no differences in δ^13^C value. We speculate that this may probably be related to the lack of sufficient food intake for a specific time, which directly affects the nitrogen isotopic ratio [31,32,33,34,35] but without changes in diet composition reflected by the values of δ^13^C [17]. Stable isotopes from the diet are reflected in the tissues of consumers and in body excretions after a metabolic process. However, this transition is not direct and goes through an isotopic fractionation [20,21,60]. Several studies have reported that fasting influences δ^15^N value [31,35,61], where tissues after a period of dietary stress are enriched in ^15^N, and thus, excretions will be relatively enriched in ^14^N, so that the decrease in the ratio between ^15^N and ^14^N in faeces leads to reduced δ^15^N values [30,31,35]. 

The contrasting effort in male–male competition between stags in both types of populations is clearly explained by the mating strategies, resembling a *quasi*-naturally experimental situation to study the different use of stored reserves by red deer males during the rut. In HC estates, with a more balanced sex ratio and many rival males, competition for mating opportunities is notably more intense than in female-biased LC estates [10,11,12,13,43]. Competing males must defend territories or harems to achieve reproductive success, while in LC populations, matings occur under a scramble framework with little competition among males [6,7,10,11,12,13,43]. The competitive situation in HC leads to the fasting of males during the rut, and hence the high energy expenditure may culminate in the deterioration of body condition and energy being produced directly through the catabolism of their tissues [34,55]. When the body’s tissues are used to produce energy, isotopic fractionation processes are repeated using the body’s reserves. This leads to more δ^15^N enriched tissues [31] and a higher proportion of ^14^N excretion in faeces [34,62]. When the mating season is over, male red deer must intake approximately twice as many kJ per day to restore the body reserves lost during the rut to recover body condition and be prepared to start the costly renewing of antlers [10,50,63]. 

We also found that in both types of population (HC and LC) parasitised individuals present lower δ^15^N values than non-parasitised ones. Parasites affect their host in various ways, affecting the immune response and causing behavioural changes [38]. They also negatively affect the host’s body condition [37,37]. It has been shown that when an individual is suffering from a disease, their δ^15^N values in tissues tend to increase [39,40], so it should be considered that within these populations, there are other factors that further differentiate the isotopic values of nitrogen, as an individual that is parasitised also has a different metabolism. Being parasitised creates isotopic variations, in turn influencing the physiological processes at the base of isotopic fractionation and the distribution of stable isotopes in the body [34]. In accordance with what occurs between individuals in populations where the competition is high and where it is low, this result supports the fact that individuals in poor body conditions show lower δ^15^N values in faeces than individuals in better body conditions.

This work may be considered as a basis to further explore the usefulness of stable isotopes and what other information they can provide despite several limitations. The δ^13^C values in faeces reflect the diet over a relatively short period of time [17], and the samples used in this study were collected during an autumn-winter period. To gain knowledge of diet fluctuations using faeces δ^13^C values, it would be necessary to collect them with continuity over time to be able to estimate changes in diet. Future studies will investigate how changes in diet composition and quality can impact wild ungulate metabolism and condition, considering both females and males to have a wide overview. It is already known that tissues and body excretions reflect the diet of the consumer an indirect relationship through isotopic fractionation [54], but even nutritional stress responses and compromised health influence the physiological processes at the base of isotopic fractionation and the distribution of stable isotopes in the body of individuals. The use of this technique in wildlife studies may be further enhanced with a larger sample size over different years. In addition to intrasexual competition, there may be other reasons why δ^15^N and δ^13^C distributions differ among red deer individuals. As a part of future work, it will be necessary to consider the factors that influence the red deer’s physical condition, its nitrogen and carbon isotopic value in faeces, such as social factors (i.e., population density and functional sex ratio), ecological variables (food availability, browsing degree) and population management practices (supplementary food).

## 5. Conclusions

-There are no differences in carbon isotopic ratio (δ^13^C) between male Iberian red deer from different intrasexual competition populations due to similar diet composition in both scenarios.-Nitrogen isotopic ratio (δ^15^N) is lower in individuals from high-intrasexual competition populations rather than low-competition populations.-This work highlights the different use of proteins by male Iberian red deer under different intrasexual competition levels in the population.-Isotopic analysis is a relevant tool for studying individual and populational variations in the intrasexual competition level with implications in evolutionary ecology and population management.

## Figures and Tables

**Figure 1 animals-13-02397-f001:**
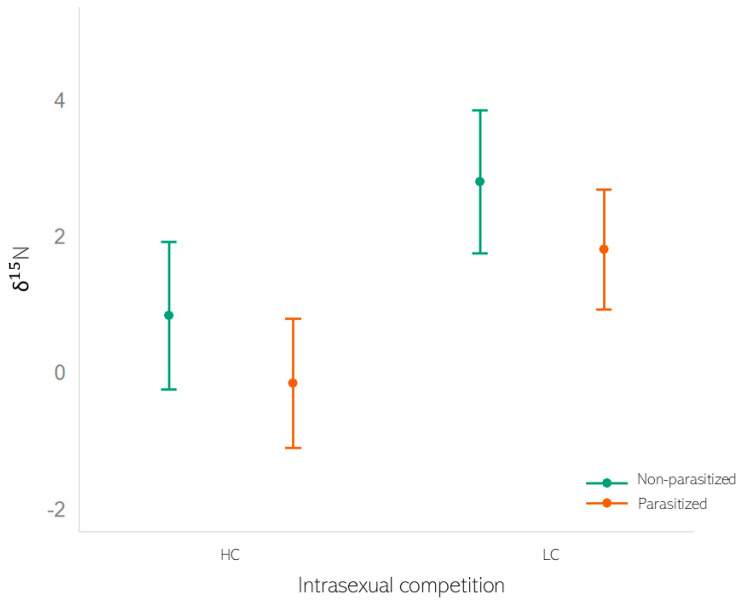
Predictions derived from LMM with logit as the link function of δ^15^N against the level of intrasexual competition in the population of Iberian red deer (HC vs. LC) for parasitised (orange) and non-parasitised (green) individuals. The error bar represents the mean, and 95% C.I. Graphic was generated using the generic function plot in R.

**Table 1 animals-13-02397-t001:** Final LMM model with logit as link function to explore the effect of the type of population (HC vs. LC) and the presence of parasites (yes vs. no parasitised) on the δ^13^C. As fixed factor δ^15^N, age, age^2^, antler length, body length, thoracic perimeter, and the dark ventral patch size. The table also shows variance and standard deviance (S.D.) of the random effects (population).

	Estimate ± S.E.	*t*-Value	*p*-Value
*Fixed factors*			
Intercept	−25.663 ± 0.702		
δ^15^N	0.337 ± 0.234	1.439	0.155
Population (LC)	−1.042 ± 0.657	−1.586	0.126
Parasitised (Yes)	0.365 ± 0.464	0.788	0.434
Age	−0.155 ± 0.892	−0.173	0.863
Age^2^	−0.165 ± 0.866	−0.190	0.850
Antler length	0.076 ± 0.262	0.289	0.774
Dark ventral patch size	−0.069 ± 0.216	−0.319	0.751
Thoracic Perimeter	−0.044 ± 0.211	−0.207	0.837
Body length	0.025 ± 0.261	0.094	0.925
*Random factor:*			
*Month*: Variance ± S.D.: 1.063 ± 1.031; *p*-value = 0.1237			
*Residual*: 1.286 ± 1.134			
$R_{m}^{2}$ = 9.4; $R_{c}^{2}$ = 50.4			
VIF < 2.44			

**Table 2 animals-13-02397-t002:** Final LMM model with logit as link function to explore the effect of the type of population (LC vs. HC) and the presence of parasites (yes vs. no parasitised) on the δ^15^N. As fixed factor δ^13^C, age, age^2^, antler length, body length, thoracic perimeter, and the dark ventral patch size. Significant effects are shown in bold (*p*-value < 0.05). The table also shows variance and standard deviance (S.D.) of the random effect (population).

	Estimate ± S.E.	*t*-Value	*p*-Value
*Fixed factors*			
Intercept	0.996 ± 0.524		
δ^13^C	0.199 ± 0.158	1.257	0.214
**Population** (LC)	1.961 ± 0.474	4.139	**<0.001**
**Parasitised** (Yes)	−0.996 ± 0.3779	−2.623	**0.011**
Age	−1.056 ± 0.752	−1.404	0.165
Age^2^	1.234 ± 0.725	1.702	0.094
Antler length	−0.025 ± 0.226	−0.109	0.914
Dark ventral patch size	−0.140 ± 0.186	−0.753	0.455
Thoracic Perimeter	0.177 ± 0.180	0.980	0.331
Body length	−0.214 ± 0223	−0.961	0.340
*Random factor*:			
*Month*: Variance ± S.D.: 0.481 ± 0.694; *p*-value = **0.048**			
*Residual*: 0.964 ± 0.982			
$R_{m}^{2}$ = 44.2; $R_{c}^{2}$ = 62.8			
VIF < 2.451			

## Data Availability

Not applicable.

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
