# Peer review of "Is the Intrasexual Competition in Male Red Deer Reflected in the Ratio of Stable Isotopes of Carbon and Nitrogen in Faeces?"

_animals, 2023, doi:10.3390/ani13142397_

Round 1

Reviewer 1 Report

The manuscript shows a preliminary approach to a question that should be of interest to a wide variety of scientists and deserves to be published.

Quite at the beginning of the manuscript, the authors talk about stress situations (L13). I would suggest avoiding this or being more precise. After reading the whole document, it is not clear to me; does it refers to “normal” endocrinological stress? nutritional stress (probably)? parasitological stress (even possible according to the analyses)? Please be accurate, or readers just reading the abstract may misunderstand the results.

Please check carefully some formal aspects. The references in brackets are quite often wrongly written (spaces between numbers or not, spaces in the middle of the numbers, etc). Similar problems with the isotopes: delta is sometimes followed by a space, delta is missing some other times (see, for example, the paragraph starting in L242). Check also the subscripts in the notation of chemical compounds (CO2, N2… L154-156, for example).

Please, indicate how the DPV was measured. It seems to me that this variable should be measured in cm2 (area), not in cm (length? which length?).

In the methods, paragraph starting in L134, there is the first information about parasitological analyses. There is no information in the introduction about how this may influence the research question. Please add it.

It is also not clear how parasites are used in the analyses. Apparently, all is summarised in a binary variable parasitised? yes/no. Please, explain some more about it. It looks to me that an animal with a low load of one of the 5 studied parasites will go to the YES category exactly the same as another individual with a high load for the 5 of them, which doesn´t make much sense.

Please, indicate the size as well of each “parasitised” category. I would expect very few individuals parasite-free, so the two categories may be quite imbalanced and have no homogenous variance. In that case, that must be considered in the models.

I´m not much sure if the z-transformation of Age2 is reasonable since this variable has a different, not-normal distribution. Since I don´t know if that´s statistically correct, I would appreciate some reference supporting this procedure.

L179. Correct the repeated text.

The tables show the full models, while the text says that the models were solved in a backwards procedure. Both options are reasonable, I would say, but please make methods and results match.

I kept the most important concern for the end. The main concept of the paper is the comparison of individuals under high and low intraspecific competition. That´s investigated through animals coming from fenced (HC) and not fenced game estates (LC). Differences between these two groups are found just for N15 but not for C13. With such a setting, it can be just concluded that intraspecific competition MAY BE the factor behind that difference. But obviously, there may be others. Any management practice creating measurable differences between both models may be the reason behind the results. Differences between both models, in carrying capacity, average home ranges, functional sex ratio, grazing competition, availability of supplementary feeding, poaching probability, etc etc, would be arguable as potential explanations. I agree with the authors that intrasexual competition can probably be the reason, but other possibilities should be identified, evaluated and discarded (if possible).

Author Response

REVIEWER 1

The manuscript shows a preliminary approach to a question that should be of interest to a wide variety of scientists and deserves to be published.

Quite at the beginning of the manuscript, the authors talk about stress situations (L13). I would suggest avoiding this or being more precise. After reading the whole document, it is not clear to me; does it refers to “normal” endocrinological stress? nutritional stress (probably)? parasitological stress (even possible according to the analyses)? Please be accurate, or readers just reading the abstract may misunderstand the results. >>> Thanks for your comments and suggestion. We agree with you that we had not been vey accurate along the manuscript. In this case, we would like to refer to nutritional stress, so we have made appropriate changes throughout the manuscript regarding this issue.

Please carefully check some formal aspects. The references in brackets are quite often wrongly written (spaces between numbers or not, spaces in the middle of the numbers, etc). Similar problems with the isotopes: delta is sometimes followed by a space, delta is missing some other times (see, for example, the paragraph starting in L242). Also check the subscripts in the notation of chemical compounds (CO2, N2… L154-156, for example). >>> Thank you for pointing out these oversights. We have checked and removed spaces where they should not be. We also checked the slopes of the chemical compounds. In the case of L242 and in other cases where δ does not appear, we are referring to the single isotopes (14N and 15N) not the ratio between them that could be expressed as δ15N.

Please, indicate how the DPV was measured. It seems to me that this variable should be measured in cm2 (area), not in cm (length? which length?). >>> The DVP was measured as the maximum length of the dark surface on the ventral zone following previous methodology (de la Peña et al., 2020 in Int. Zool. & Sci. Nat.; Carranza et al., 2020 Sci. Rep.) We changed this information in the methods section, and we also added the main reference of this topic: Carranza et al., 2020.

In the methods, paragraph starting in L134, there is the first information about parasitological analyses. There is no information in the introduction about how this may influence the research question. Please add it. >>> Thanks. We agree with you, so we have added a paragraph in the introduction linking the parasites presence effect to the stable isotopes distribution in the organism.

  1. 77-82: “In this context, the presence of parasites in organisms is of relevance, since it is well known that parasite have a detrimental effect on the body condition [36,37], immune response and behaviour of their hosts [38]. Parasitisation leads to isotopic variations that influence physiological processes related to isotope fractionation and the distribution of stable isotopes in the organism [34]. Previous studies have shown that parasitism in-creases δ15N values in tissues [39,40].”

It is also not clear how parasites are used in the analyses. Apparently, all is summarised in a binary variable parasitised? yes/no. Please, explain some more about it. It looks to me that an animal with a low load of one of the 5 studied parasites will go to the YES category the same as another individual with a high load for the 5 of them, which doesn´t make much sense. >>> The individuals included in this study were examined by coprology for the presence or absence of gastrointestinal nematodes, trematodes, cestodes, coccidian oocysts, and bronchopulmonary nematode larvae. However, only gastrointestinal parasites (Elaphostrongylus cervi) and bronchopulmonary parasites (Strongylida spp.) were found to be infected. Therefore, the presence/absence of parasite load refers to both parasite groups. We decided not to consider the parasite load, and to take into account just prevalence because we were interested in studying the susceptibility of individuals to parasitism as a proxy for their immune function (please, see de la Peña et al., 2020 Int. Zool.). We agree that this information was not well specified in the manuscript, so we have revised this part of the methodology.

  1. 147-152: “The individuals included in this study were examined for the presence or absence (i.e., prevalence) of gastrointestinal nematodes, trematodes, cestodes, coccidian oocysts, and bronchopulmonary nematode larvae. However, only gastrointestinal parasites (Elaphostrongylus cervi) and bronchopulmonary parasites (Strongylida spp.) were found to be infected. Therefore, the presence/absence of parasite prevalence refers to both parasite groups.”

Please, indicate the size as well of each “parasitised” category. I would expect very few individuals parasite-free, so the two categories may be quite imbalanced and have no homogenous variance. In that case, that must be considered in the models. >>>

We agree with you that the two groups of our categorical variable being parasitized or not is not balanced:

  • Parasitised 50 individuals (30 individuals from HC estates vs 20 individuals from LC estates)
  • Non-parasitised 17 individuals (12 individuals from HC estates vs 5 individuals from LC estates)

However, we decided to use LMMs because it has the potential to deal with unbalanced and incomplete data sets in contrast to the ANOVA test. To ensure that the results of the models built for this paper were correct, we checked the normality of the residuals by means of the Shapiro test (obtaining p-values > 0.05 in both models) and visual exploration in QQplots, the presence of outliers and influential points, as well as the singularity of the model (by using the library performance in R). We also checked for the presence of multicollinearity using VIF. Therefore, we believe that the structure of the random and fixed effects in both models of the paper meet the assumptions of the LMMs to obtain reliable data regardless the unbalanced structure of the variable parasitised (YES/NO).

I´m not much sure if the z-transformation of Age2 is reasonable since this variable has a different, not-normal distribution. Since I don´t know if that´s statistically correct, I would appreciate some reference supporting this procedure. >>> By z-transformation we mean scaling variables using the scale () function in R studio. With this function we aim to be able to compare data that have not been measured in the same way (normalization of the data set using the mean value and standard deviation). To obtain reliable results, we normalize variables that are numerical regardless of the distribution it follows (quadratic in the case of Age 2).

Here, the information about the R package:

https://www.rdocumentation.org/packages/base/versions/3.6.2/topics/scale

Here we also leave a discussion on how to apply this type of standardization to quadratic variables:

https://stats.stackexchange.com/questions/264146/standardizing-quadratic-variables-in-linear-model

L179. Correct the repeated text. >>> OK, changed.

The tables show the full models, while the text says that the models were solved in a backwards procedure. Both options are reasonable, I would say, but please make methods and results match. >>> You are right. The models that we show in tables are the final models, not the full models. We changed this, thank you for your help.

 I kept the most important concern for the end. The main concept of the paper is the comparison of individuals under high and low intraspecific competition. That´s investigated through animals coming from fenced (HC) and not fenced game estates (LC). Differences between these two groups are found just for N15 but not for C13. With such a setting, it can be just concluded that intraspecific competition MAY BE the factor behind that difference. But obviously, there may be others. Any management practice creating measurable differences between both models may be the reason behind the results. Differences between both models, in carrying capacity, average home ranges, functional sex ratio, grazing competition, availability of supplementary feeding, poaching probability, etc etc, would be arguable as potential explanations. I agree with the authors that intrasexual competition can probably be the reason, but other possibilities should be identified, evaluated, and discarded (if possible). >>> We fully agree with your comment. In this paper, we intend to give a first approximation of the information on the isotopic value of nitrogen and carbon in faeces in males under different mating competition scenarios. But this paper sheds light on several studies that explore the relationship with diet composition, including browsing intensity and supplementary feeding, and explore differences between sexes, and ages to know the factors modulating nitrogen and carbon isotopic values. We have expanded in the discussion the paragraph where these possibilities are discussed, hoping to improve this section and the implications of the work satisfactorily.

  1. 321-326: “In addition to intrasexual competition, there may be other reasons why δ15N and δ13C distributions differ among red deer individuals. As part of future work, it will be necessary to consider the factors that influence the red deer's physical condition, its nitrogen and carbon isotopic value in faeces, such as social factors (i.e., population density and functional sex ratio), ecological variables (food availability, browsing degree), and population management practices (supplementary food).”

Reviewer 2 Report

The manuscript is worth publishing after minor additions/explanations/corrections.

The text is well written and the results are clear and interesting.

I suggest some small changes in specific points:

Line 118: Please explain better here why in fenced and unfenced estates male intrasexual competition is different. We can find an explanation in the paragraph on results but you should clarify also here.

Line 129: body length: Is it the “head-trunk length” (excluding tail)? Please clarify.

Lines 234-235: Please add a title, Apollonio et al. 2020 “Capital-income breeding in male ungulates…”

Line 254: please correct “cuasi” into “quasi” (Latin).

Line 287: “Future studies will study…”. Please avoid a repetition

Author Response

REVIEWER 2

The manuscript is worth publishing after minor additions/explanations/corrections.

The text is well written, and the results are clear and interesting. I suggest some small changes in specific points:

Line 118: Please explain better here why in fenced and unfenced estates male intrasexual competition is different. We can find an explanation in the paragraph on results, but you should clarify also here. >>> OK, thanks. We add some information in this section to ensure that it is clear to the reader the differences between unfenced and fenced farms in terms of the intrasexual competition situation and where it comes from. We hope that this change will be satisfactory, but we are willing to make any necessary changes in future revisions.

  1. 127-132: “This results in a sex ratio bias to females. In this scenario, males can mate without the need to compete with other rivals [10,43-44]. On the contrary, fenced estates maintain more equilibrated age and sex population structures that allow intrasexual competition to operate more naturally [43]. Thus, the females’ availability is lower at these estates and males must compete to mate with them [10,43-44]. There is no difference between un-fenced and fenced estates in terms of population density due to these two management situations.”

Line 129: body length: Is it the “head-trunk length” (excluding tail)? Please clarify. >>> We measured body head-tail-length. We included this information in the manuscript. Thanks.

Lines 234-235: Please add a title, Apollonio et al. 2020 “Capital-income breeding in male ungulates…” >>> OK, thanks. We already added this reference.

(1)       Apollonio, M.; Merli, E.; Chirichella, R.; Pokorny, B.; Alagić, A.; Flajšman, K.; Stephens, P. A. Capital-Income breeding in male ungulates: causes and consequences of strategy differences among species. Front. Ecol. Evol. 2020, 8.

Line 254: please correct “cuasi” into “quasi” (Latin). >>> OK, thanks, changed.

Line 287: “Future studies will study…”. Please avoid a repetition. >>> OK, thanks, changed.

Reviewer 3 Report

The paper describes how the use of carbon and nitrogen isotopes can be useful in to study of the ecology of ungulates and in particular, the feeding behavior of male deer during the reproductive season, in two different management situations: fenced areas and open areas, with a density of males, considered different. The work is interesting but must be resubmitted with data from censuses and the relationships between the sexes and between the age groups, in the two areas (managements)  of study. The reference data, for supporting the hypothesis,  in the text, are publications not coinciding with the period of study, and on several areas, therefore difficult to associate. In addition, the availability of food should and habitat characteristics, also be analyzed (also through indexes of browsing)  of the two areas that I expect, as described, may undergo different foraging pressures, depending on the different management and density. In my opinion, therefore, there is no robust ecological framework in terms of animals censused and habitats, elements that could then also be included in any statistical calculations. Also, the work is given much weight to the presence of the parasites but this part is not mentioned in the summary and above all is not described in accurate terms in terms of the presence-prevalence. The work is therefore very lacking in terms of materials and methods and also of accessory results

Author Response

REVIEWER 3

The paper describes how the use of carbon and nitrogen isotopes can be useful into study of the ecology of ungulates and in particular, the feeding behaviour of male deer during the reproductive season, in two different management situations: fenced areas and open areas, with a density of males, considered different. The work is interesting but must be resubmitted with data from censuses and the relationships between the sexes and between the age groups, in the two areas (managements) of study. The reference data, for supporting the hypothesis, in the text, are publications not coinciding with the period of study, and on several areas, therefore difficult to associate. In addition, the availability of food should and habitat characteristics, also be analysed (also through indexes of browsing) of the two areas that I expect, as described, may undergo different foraging pressures, depending on the different management and density.

In my opinion, therefore, there is no robust ecological framework in terms of animals censused and habitats, elements that could then also be included in any statistical calculations.  Also, the work is given much weight to the presence of the parasites, but this part is not mentioned in the summary and above all is not described in accurate terms in terms of the presence-prevalence. The work is therefore very lacking in terms of materials and methods and also of accessory results.

>>> Thanks for your comments and suggestions. We understand your concern about the need to include variables related to the age and sex structure of the different populations, and indicators of red deer foraging behaviour to improve this study. Unfortunately, we are unable to retrieve this information as data are from 2017 and 2018. Furthermore, our main goal is to explore the differences in the isotopic value of nitrogen and carbon in two different scenarios of intrasexual competition in male red deer from the point of view of evolutionary ecology. Previous works published by our research group focus on the same study area (many of these hunting estates have been visited for more than 20 years), so there is a precise coincidence with respect to the cited bibliography. For instance, Carranza et al. (2020; in PLOS ONE on the DVP) included data between 2005 and 2019 from the study area and looked at differences between both types of populations (HC and LC), and there are several other works also including the area and period. We have shown that male red deer in unfenced (LC) and fenced (HC) areas suffer different intrasexual competition pressure due to hunting management, which affects population structure and sexes regardless of the year sampled, because the contrasting management differences are maintained throughout the years in these two types of hunting estates. In previous works we demonstrate that due to this different degree of mating competition, the development of sexual characteristics by males is different (this information is included in the Discussion L. 278-293). But also, we have used the two management situations (and the associated level of male-male competition) as a categorical factor with two levels rather than a quantitative variable, so we are confident that we have no error in this classification. Therefore, we consider that the ecological framework of the work is adequate and robust for the objective we are proposing. To improve the reader's understanding of this part we have added this information in the methodology section (L. 127-132). Please do not hesitate to contact us if you think that there is a better way of capturing this information in a way that is reflected and understandable to the reader, and we would be very grateful to improve the manuscript.

This work suggests a new approach to be considered for future work to study the physiological effect of the social and ecological environment, including red deer population density, sex ratio, food, and water availability, among others, considering both sex and age of the red deer. But from a humble and conservative point of view, we propose the use of stable isotopes as an indicator of the mate competition level in the population.

Taking into account these considerations and comments, we have expanded in the discussion the paragraph on limitations of the work and future studies:

  1. 321-326: “In addition to intrasexual competition, there may be other reasons why δ15N and δ13C distributions differ among red deer individuals. As part of future work, it will be necessary to consider the factors that influence the red deer's physical condition, its nitrogen and carbon isotopic value in faeces, such as social factors (i.e., population density and functional sex ratio), ecological variables (food availability, browsing degree), and population management practices (supplementary food).”

Regarding the presence of parasites, we have added information in the introduction on the implications of being parasitized or not in the context of the work, and also in the methodology, we have described in a more accurate way the number of parasitized and non-parasitized individuals in the two contexts of intrasexual competition. If you think it would be appropriate for us to add any further information, we would be happy to make any additional changes.

Round 2

Reviewer 3 Report

the authors responded only partially to my doubts and observations, which I had raised previously; in fact, they reflect, in my opinion, aspects that should be discussed and better described, which also extends to the part of statistical analysis and its robustness, which the authors should better describe

The description of the nine areas of study is not present; only an average surface value is described; the surface would be described, by type of population, also with other parameters such as minimum and maximum; I also suggest reporting the % habitat coverage for the two study areas also in terms of % of the main habitats

The relationship between females and males are not described for the two populations and neither are bibliographic values mentioned; in my opinion, it would be necessary to put the citations, also to report in the text data of density and sex ratio, although previously described in other works

Information on the number of parasitic animals in the two types of population is lacking in the materials and methods; in general, the number of animals with and without parasites in both populations and tables is not indicated. The number of animals measured and examined is 78 the number of feces analyzed is 67: it is necessary to explain how these groups are consistent with each other and the application of the model on which data was made, or what number there was, from the data shown, is not clear

In general, in the results, there are no even descriptive representations of the analyzed factors broken down by types of population and level of parasitization, I suggest building some graphs ( or tables) with the mean and minimum, and maximum values or the variability coefficients of the fixed factors then considered in the models, such as age, thoracic perimeter....

the results also fail to indicate the values of N , although they did not result significantly different between groups, and generally, I suggest showing the values observed for N, C between groups.

The statistical part, of the results,  does not appear clear in terms of the models presented; the materials and methods describe how non-significant fixed effects have been removed to reduce the risk of over-parameterization, then in the model of Table 1 appear all the factors described; can this affect the effect of the factors, considered subsequently significant? Is the number of samples sufficient to build the exposed model, made of 8 fixed factors, 1 random effect, and the intercept (n=67  samples)? you  should describe why only one model was shown  and if You have analysed also the correlation between the fixed factors

In Figure 1 it would be better to write the caption in terms of the described variable, moreover, the lengths of the bars seem strangely similar for the groups, could you check and indicate the numerical values also in the text? 

Author Response

REVIEWER 3 – Second round

The authors responded only partially to my doubts and observations, which I had raised previously; in fact, they reflect, in my opinion, aspects that should be discussed and better described, which also extends to the part of statistical analysis and its robustness, which the authors should better describe. >>> Thanks for your comments and suggestions. We have tried to consider all the changes proposed by you in this revised version to improve the description of the methodology, statistical analysis, and discussion of the results. We hope that your concerns have been satisfactorily addressed and this paper is ready for publication. If you think it would be appropriate for us to add any further information, we would be happy to make any additional changes to improve the manuscript.

The description of the nine areas of study is not present; only an average surface value is described; the surface would be described, by type of population, also with other parameters such as minimum and maximum; I also suggest reporting the % habitat coverage for the two study areas also in terms of % of the main habitats. >>> Thank you for your comment. We have added information on the average area of the two types of populations and information on the minimum and maximum area in each case. We have also included specific information on the study areas and average density of Iberian red deer.

The nine study populations presented are located within a specific study area in the south-west of the Iberian Peninsula. It is an area where the vegetation present is similar, with a predominance of open holm oak forests and other Mediterranean species such as Cistus spp., Erica spp., Arbutus unedo, Phyllirea spp., Genista hirsuta, Lavandula spp. and Olea europaea. Initially we thought that due to this homogeneity in the landscape across our study area it would not be necessary to describe each of the study populations in an exhaustive manner, as this could overwhelm readers and would not provide the information necessary to address the main objective of the article and the discussion of the results. In previous work of our research group based on identifying differences in the development of sexual traits in male red deer in the same study area between populations with high and low intrasexual competition, information on habitat quality was collected considering changes in the composition of the shrub vegetation mosaic, following the index described by Pérez-González & Carranza, 2009. It is possible that there have been small changes over the years in this index, but relative differences between population types are expected to be maintained. In any case, we have described in detail the plant formations and species that can be found in the study area in the methodology.

Please check L. 107-144 to see all this changes: “We investigated male Iberian red deer harvested in hunting estates in southwestern Spain (Andalusia and Extremadura regions). This study was carried out in nine populations concretely located in Sierra Morena (Province of Córdoba, UTM 37º58′ N, 5º05′ W) and Sierra San Pedro (Province of Cáceres, UTM 39º19′ N 6º42′ W). The vegetation present is similar throughout the study zone. It is characterized by mountain ranges covered by open oak agroforestry woodland, known as dehesa, accompanied by Mediterranean species such as Cistus spp., Erica spp., Arbutus unedo, Phyllirea spp., Genista hirsuta, Lavandula spp. and Olea europaea [12-13,40]. Each hunting estate constitutes a different population because of the existing natural (i.e., mountains, geographical distance) and artificial (i.e., fences) barriers between them. The average estates area in our study area was 2347 ha, being a minimum area of 503.07 ha and the maximum area 9223.18 ha. Two different management regimes apply to hunting estates in our study area. In one regime, estates are fenced with stock-proof wire mesh and red deer are not allowed to move across nearby estates, while in the other regime, estates are unfenced [43,44]. In unfenced estates, the free movement of animals from one estate to another means that as many as possible male red deer are hunted in each estate to prevent them from being hunted in neighbouring estates [44]. This results in a sex ratio bias toward females. In this scenario, males can mate without the need to compete with other rivals [10,43-44]. On the contrary, fenced estates maintain more equilibrated age and sex population structures that allow intrasexual competition to operate more naturally [43]. In unfenced estates, the average area was 3948.44 ha while in fenced estates the average area was 1074.28 ha. Females’ availability is lower at these estates and males must compete to mate with them [10,43-44]. There is no difference between unfenced and fenced estates in terms of population density due to these two management situations [44], the density of individuals in both types of populations does not differ [44] being around 0.3 individual/ha (0.1-1.0 indiv. /ha) [44]. However, they show contrasting scenarios of male intrasexual competition for mates. Referred to as high competition (HC) in fenced estates and low competition (LC) in unfenced estates in several previous studies [9,12-13,45,10,11]. Specifically, previous work has reported that the sex ratio in open hunting estates was mean ± SE = 4.76 ± 0.40 (females/males), while in fenced hunting estates, the sex ratio was mean ± SE = 1.58 ± 0.13 (females/males) [43-44]. Henceforth, we will refer to fenced estate populations as having a high mate competition level (HC) and to unfenced estate populations as having a low mate competition level (LC).”

The relationship between females and males is not described for the two populations and neither are bibliographic values mentioned; in my opinion, it would be necessary to put the citations, also to report in the text data of density and sex ratio, although previously described in other works. >>> OK, we have included specific information on the sex ratio of unfenced vs. fenced estates based on previous studies. Please, check L. 139-142. “Specifically, previous work has reported that the sex ratio in open hunting estates was mean ± SE = 4.76 ± 0.40 (females/males), while in fenced hunting estates, the sex ratio was mean ± SE = 1.58 ± 0.13 (females/males) [39].”

In addition to having cited throughout the work several previous works where the implications of the two types of hunting management have been demonstrated in the sex and age structure of the red deer population in our study area, we would like to emphasize the use of genetics techniques in the following scientific articles that support these differences in the sex ratio that we use:

Pérez-González, J., Mateos, C & Carranza, J. 2009. Polygyny can increase rather than decrease genetic diversity contributed by males relative to females: evidence from red deer. Molecular Ecology 18, 1591-1600.

Pérez-González, J.  & Carranza, J. (2011). Female aggregation interacts with population structure to influence the degree of polygyny in red deer. Animal Behaviour 82, 957-970.

Information on the number of parasitic animals in the two types of population is lacking in the materials and methods; in general, the number of animals with and without parasites in both populations and tables is not indicated. The number of animals measured and examined is 78 the number of faeces analyzed is 67: it is necessary to explain how these groups are consistent with each other and the application of the model on which data was made, or what number there was, from the data shown, is not clear. >>> We included this information in the previous revision as part of the response to the reviewers, but we did not include it in the methodology. According to your comments and suggestions, in this version, we have added the information on parasitized and non-parasitized individuals for each type of population (LC vs HC) in the methodology at L.163-165. “A total of forty-two males were sampled from forty-six individuals in HC populations for this study. From LC populations, samples were collected from twenty-five males out of thirty-two.”

Information on the number of parasitized and non-parasitized individuals for each type of population is included in Table S1 as Supplementary material including the rest of the descriptive statistics for the variables considered in the analyses.

In general, in the results, there are no even descriptive representations of the analyzed factors broken down by types of population and level of parasitation, I suggest building some graphs (or tables) with the mean and minimum, and maximum values or the variability coefficients of the fixed factors then considered in the models, such as age, thoracic perimeter.... >>> As part of the Supplementary Material, we included descriptive statistics for each variable used in the main models per type of population (HC vs LC).

Table S1: Mean ± SD, minimum and maximum values of age, antler length (AL, cm), body length (BL, in cm), thoracic perimeter (TP, in cm), and the dark ventral patch size (DVP, in cm) per type of population (HC vs LC) as covariables included in LMM1 and LMM2. Percentage of parasitized and non-parasitized individuals in both HC and LC populations as a factor of LMM1 and LMM2.

Variable

Age

AL

BL

TP

DVP

δ13C

δ15N

Parasitized

Mean ± SD

(Minimum / Maximum values)

Yes

No

HC (N= 46*)

3.37 ± 1.68

(2/8)

57.7 ± 9.58

(36.8/78.5)    

181 ± 10.8

(154/201)

119 ± 6.16

(100/130)

36.8 ± 25.4

(0/73)

- 26.3 ± 1.57

(-28.8/-21.7)  

0.511 ± 1.28

(-2.17/2.87)

71.42 %

28.57 %

LC (N = 32*)

2.97 ± 1.52

(2/9)

52.3 ± 11.60

(36.4/82.6)

175 ± 12.1

(154/200) 

114 ± 8.18

(100/132)  

28.3 ± 20.4

(5/67)

-26.1 ± 1.30

(-28.9/-23.7)

2.25 ± 1.40

(-0.49/4.76)

80 %

20 %

* Faecal samples from 42 individuals of HC and 25 individuals from LC populations

The results also fail to indicate the values of N, although they did not result significantly different between groups, and generally, I suggest showing the values observed for N, C between groups. >>> OK, we have added this information also in Table S1.

The statistical part, of the results, does not appear clear in terms of the models presented; the materials and methods describe how non-significant fixed effects have been removed to reduce the risk of over-parameterization, then in the model of Table 1 appear all the factors described; can this affect the effect of the factors, considered subsequently significant? Is the number of samples sufficient to build the exposed model, made of 8 fixed factors, 1 random effect, and the intercept (n=67 samples)? you should describe why only one model was shown and if You have also analyzed the correlation between the fixed factors. >>> We have fixed an error in the writing of the methods in the body of the text, as to reduce the risk of over-parameterization we removed non-significant interactions. Our final decision was to include all the morphological variables of individuals since previous work has shown that they are influenced by intrasexual competition levels (HC vs LC). We think that the models are correctly specified as there is no convergence problem, and random effects contribute variance in both cases. The residuals of the model were normal (tested by the Shapiro-Wilks test and visualized by Qqplots). There is also no singularity nor outliers (tested using the performance package in R Studio). The collinearity between variables was checked by calculating the variance inflation factor (VIF) using the performance package in R Studio. Next, we expose the R output so that it is possible to check that there was no collinearity between the variables included in both models (except for age and age squared):

Low Correlation         Term  VIF     VIF 95% CI Increased SE Tolerance Tolerance 95% CI scale(d13C) 1.10 [ 1.01,  2.18]         1.05      0.91     [0.46, 0.99]      Estate 1.17 [ 1.04,  1.83]         1.08      0.85     [0.55, 0.96] Parasitized 1.13 [ 1.02,  1.92]         1.06      0.88     [0.52, 0.98]   scale(AL) 2.44 [ 1.84,  3.49]         1.56      0.41     [0.29, 0.54]  scale(DVP) 1.99 [ 1.54,  2.81]         1.41      0.50     [0.36, 0.65]   scale(TP) 1.54 [ 1.25,  2.17]         1.24      0.65     [0.46, 0.80]   scale(BL) 1.79 [ 1.41,  2.53]         1.34      0.56     [0.40, 0.71] High Correlation          Term   VIF     VIF 95% CI Increased SE Tolerance Tolerance 95% CI   scale(Age) 28.97 [19.55, 43.17]         5.38      0.03     [0.02, 0.05] scale(Age^2) 26.70 [18.04, 39.78]         5.17      0.04     [0.03, 0.06]

For all these reasons, we think that the structure of the models is correct and that this part is adequately described in the current version thanks to your comments. If there are any suggestions or changes you think we should make, we are happy to improve any part of the statistical analysis.

In Figure 1 it would be better to write the caption in terms of the described variable, moreover, the lengths of the bars seem strangely similar for the groups, could you check and indicate the numerical values also in the text? >>> We have re-named the variables in Figure 1 as described above:

We use the ggeffects package in R Studio (described in the methodology) to plot the results derived from LMM1. This package computes estimated marginal means, i.e., predicted values, for the response, at the margin of specific values or levels of certain terms in the model. It generates predictions of a model by holding non-focal variables constant and varying the focal variables, in our case level of intrasexual competition in the population and non-parasitized or parasitized. Therefore, the bars are similar in each group. Figure 1 comes from the following information extracted from this package:

# Parasitized = NO Estate   | Predicted |        95% CI------------------------------------Fenced   |      0.71 | [-0.69, 2.10]Unfenced |      2.82 | [ 1.49, 4.15] # Parasitized = SI Estate   | Predicted |        95% CI------------------------------------Fenced   |     -0.26 | [-1.52, 0.99]Unfenced |      1.86 | [ 0.68, 3.03] Adjusted for:*  d13C = -26.03*   Age =   3.20*    AL =  56.01*   DVP =  33.49*    TP = 118.38*    BL = 179.06* Month = 0 (population-level)
